# A Numerical Analysis on Lateral Resistance of Pile–Bucket Foundation for Offshore Wind Turbines

Zonghao Yuan [1], Ke Zhong [1], Xiaoqiang Wang [1], Xiaodong Pan [1,*], Ben He [2], Jing Wu [1], Jiankun Zhang [1] and Shiwu Xu [1]

1   College of Civil Engineering, Zhejiang University of Technology, Hangzhou 310023, China; yuanzh@zju.edu.cn (Z.Y.); chungk_o@163.com (K.Z.); wongxq310@163.com (X.W.); zj_wujing@163.com (J.W.); zhangjk150@126.com (J.Z.); xshiwu123@sina.com (S.X.)
2   Key Laboratory for Far-Shore Wind Power Technology of Zhejiang Province, Powerchina Huadong Engineering Corporation Limited, Hangzhou 311122, China; he_b2@ecidi.com
*   Correspondence: panxd@126.com

**Abstract:** The large-diameter single pile has been widely used in marine wind turbine foundations. In order to improve its lateral bearing capacity, the suction bucket foundation (BF) can be used as a reinforcement method, which surrounds the periphery of the monopile foundation (PF), forming a new wind turbine foundation, i.e., pile–bucket foundation (P–BF). By using the general finite element software ABAQUS, in this paper, several numerical models are established to investigate the influence of pile diameter and bucket diameter on the lateral bearing capacity of large-diameter monopiles. The numerical results show that the internal force distribution along the pile shaft for the case of P–BF is similar to the case of PF. Compared to bucket height, bucket diameter has more of an effect on lateral capacity and the *p-y* curve in the P–BF. In the combined P–BF, the buckets can provide lateral bearing capacity, and the piles can provide anti-overturning moments, resulting in higher lateral bearing capacity. In this paper, the *p-y* curve in the API specification is modified based on the results from the finite element simulation. The modified *p-y* curve fits well with the results of the finite element calculation and can be used as a reference for the design of the P–BF in actual engineering.

**Keywords:** pile–bucket foundation; *p-y* curve; lateral ultimate soil resistance; soil resistance distribution; ABAQUS finite element simulation

## 1. Introduction

In the next decade, global wind-power-installed capacity will reach 680 million kW, of which 40% will be offshore wind power. The expansion of offshore wind energy will soon exceed the level of the heyday of the oil and gas industry; therefore, much energy has been invested in research and development. China's offshore wind power industry has steadily expanded under the guidance of the government. Due to the variation in water depth in different sea areas, the form of ocean wind turbine foundations also varies. In the sea areas within 30 m of water depth, gravity shallow foundations, suction bucket foundations (BF), and monopile foundations (PF) are mainly used [1,2]. The super large-diameter PF with a diameter of 4~8 m has become the main form of foundation, with the highest utilization rate of offshore windfarms, which has the advantages of fast construction and good economy.

The main considerations for offshore wind turbine foundation design are foundation weight, wind, water flow and wave, impact of floating objects, seismic, etc. The most important load is lateral force. Therefore, this paper mainly studies the lateral bearing capacity of the wind turbine foundation. At present, the *p-y* curve is the most effective nonlinear method for describing the lateral deformation and lateral bearing capacity of pile foundations, which is proposed by Matlock H. [3] based on the laterally loaded pile test in clay. Many experts and scholars, such as Zhang Y. [4], have studied the *p-y* curve by means of laboratory element tests.

Through numerical simulation of large diameter monopiles, Sun D. [5] and He B. [6] found that the existing *p-y* curve cannot be applied to large-diameter monopiles directly. Liu R. [7] et al. studied the lateral bearing capacity and settlement deformation of the pile–bucket foundation and investigated the contribution of the bucket and pile separately to the resistance of the foundation. Albusoda et al. [8] offers a nonlinear 3D analysis of pile–soil interaction, with a sequence of laboratory model tests conducted, and has developed *p-y* curves of laterally loaded finned and regular piles in multilayered sandy soil. Asgarian et al. [9] considered pile–soil interactions and obtained the ultimate strength of the platform in the non-linear pile stub case, which was close to the base case. A. Carstensen [10], based on the subgrade reaction method, allowed for modeling of a limited number of cycles and predicted the displacement of piles under cyclic lateral loading. Lv Y. et al. [11] used the ALE (Arbitrary Lagrangian–Eulerian Method) technique to study large deformations during the sinking process of the suction bucket in clay and found suction penetration resistance to be significantly lower than pressure penetration resistance. According to the model test, Andersen K. [12] put forward that increasing the buried depth of the suction bucket can improve the anti-overturning ability of the BF. Huang Z. [13] et al. analyzed the vertical distribution profile of the external friction resistance, soil pressure and axial force of the pile–tube composite foundation under the action of vertical load and obtained the degree of improvement of the vertical bearing capacity of the pile–tube composite foundation by the reinforcement of the tube body. Zhu B. et al. [14,15] and Huang M. [16] et al. studied the bearing capacity of large-diameter PF under static and cyclic loadings and then modified the *p-y* curve as recommended by the American Petroleum Institute (API) [17]. Liu W. [18] studied the P–BF under a compressive load and proposed its optimal geometrical dimensions combination.

Large-diameter piles with large stiffness demonstrate that the reaction is mainly made by the stiffness of shallow soil but that the mechanical property of shallow soil is poor. Our research orientation of designing a new foundation for wind turbines concentrate on expanding the resistance of a wider range of soil. Based on this, this paper proposes that the pile–bucket foundation (P–BF), which combines the suction bucket foundation (BF) and monopile foundation (PF) in a shallow position, enhances lateral bearing capacity.

However, until now, no studies have focused on the lateral bearing capacity of the P–BF through the *p-y* curve or have modified the *p-y* curve by the API to be applicable to practical engineering.

In this paper, the finite element software ABAQUS is used to establish the P–BF (Figure 1). Taking a large-diameter single pile with a diameter of 6 m and length of 80 m as the main body, the P–BF model is established by ABAQUS. We quantitatively analyze the bearing capacity, soil reaction, displacement distribution and *p-y* curve of three different diameter buckets. The *p-y* curve is studied to analyze the improvement in the lateral bearing capacity of the P–BF. The advantage of P–BF compared with the suction bucket is summarized.

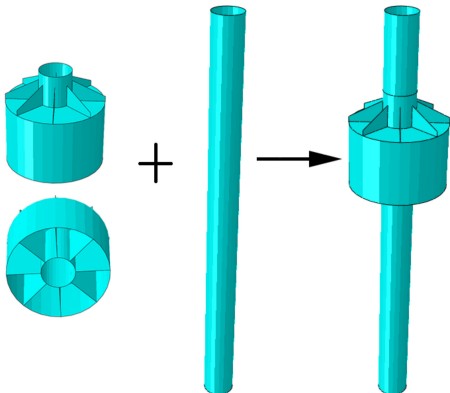

**Figure 1.** The geometry for the pile–bucket foundation; structural diagram.

## 2. Finite Element Model

### 2.1. Overview of Offshore Wind Farms

The offshore windfarm (Figure 2) is located in the Pinghu sea area of Hangzhou Bay. It has an average length of about 9 km from east to west, and 2 to 17 km from north to south. The planned sea area is about 84 km². The sea area is about 48 km², and the planned capacity is 300 MW. The center of the windfarm is about 20 km offshore. The seabed topography changes little in the total area, and the water depth is 8~12 m. The position of the underwater beach is relatively flat, the elevation is generally −7.90~−8.60 m, and the maximum slope of the seabed is less than 1°. For the engineering geology condition described above, it is suitable to PF or BF for the wind turbines.

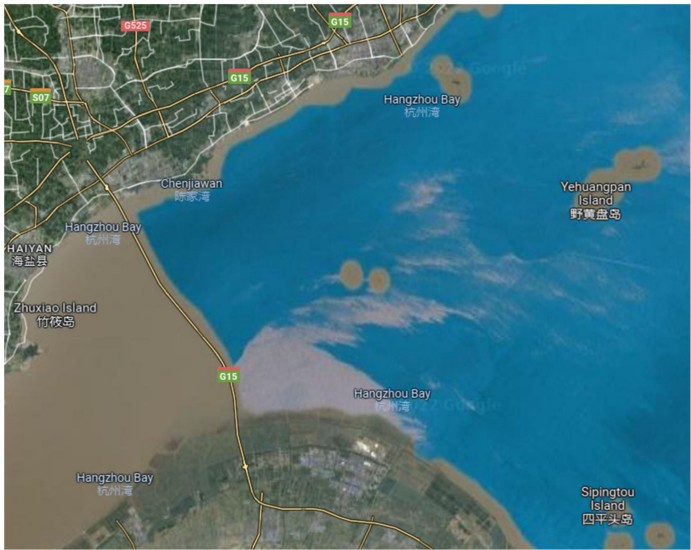

**Figure 2.** Location of the offshore windfarm.

### 2.2. Modeling of Pile Bucket Foundation

The steel (Q345C) of the foundation is modeled as an ideal elastoplastic material. It has a yield strength of 325 MPa, Young's modulus $E = 2.1 \times 10^8$ kPa, and Poisson's ratio ($v$) 0.25. The pile diameter ($d$) is chosen as 6 m, which is frequently used in practical engineering. The pile length ($l$) is 80 m (the depth embedded into the soil is 60 m). The thickness of the pile wall ($\delta$) is around 1% of the pile diameter; thus, it is 6 cm. Due to the bucket performed as a reinforcement structure, in order to save material and improve its economic benefits, the thickness of the bucket wall ($\delta$) is 2 cm. The physical parameters are summarized in Table 1.

**Table 1.** Physical parameters of the foundation.

| Yield Strength | Pile Diameter | Pile Length | Thickness of Pile Wall | Young's Modulus | Poisson's Ratio |
|---|---|---|---|---|---|
| $f_y$/MPa | $d$/m | $l$/m | $\delta$/cm | $E$/MPa | $v$ |
| 325 | 6 | 80 | 2 | $2.1 \times 10^8$ | 0.25 |

The diameter ($D$) and height ($h$) of the bucket are shown in Table 2. The width of the soil in the model is far greater than the diameter of the pile and bucket (length × width × depth = 150 × 150 × 80 m). Thus, the influence of the boundary can be neglected. The Mohr–Coulomb constitutive model (M–C model) is used. The physical parameters are shown in Table 2.

**Table 2.** Physical parameters of the soil.

| Soil Layer | | Effective Unit Weight | Young's Modulus | Friction Angle | Cohesive | Poisson's Ratio |
|---|---|---|---|---|---|---|
| | | $\gamma$/kN/m$^3$ | $E$/MPa | $\varphi$/° | $c$/kPa | $v$ |
| Surrounding | Muddy clay | 7.6 | 10 | 28.81 | 15 | 0.29 |
| Bottom | Silty clay | 8.2 | 20 | 35.75 | 8 | 0.32 |

*2.3. Meshing and Interaction*

The element type of the soil adopts the eight-node linear brick reduced integration element (C3D8R) and enhances the hourglass stiffness. This element has high accuracy for the displacement and does not cause shear locking. The element of the pile and the bucket adopt the four-node doubly curved shell (S4), and the finite membrane strain is adopted. The shell is more suitable for analyzing the characteristics of the large-diameter steel pipe pile. The details are shown in Figure 3.

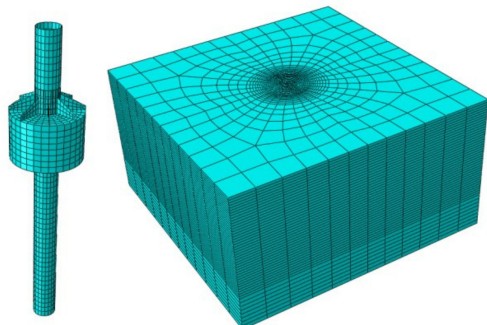

**Figure 3.** Meshing of parts.

Assembling the pile and the bucket, the inner and outer surfaces of the pile and bucket follow the Mohr–Coulomb friction law. Contact pair is characterized by small sliding and surface-to-surface contact. The concept of "elastic slip deformation" is used in the friction model of the ABAQUS contact surface, allowing for a small amount of relative slip deformation when the pile–soil contact surface is bonded together. This is because under the action of horizontal load, the pile will slide and deform with the soil when it rotates or flexes. The normal contact behavior is selected as the hard contact, and the tangential behavior is selected as the penalty, which both ensure that the pile–soil interaction model can better reflect the actual situation of the pile foundation's working properties. The friction coefficient is determined by the friction angle of the soil, according to $\tan(0.75\varphi)$ [19]. The same is applied to the bucket lower surface of the cover. The master surface is the pile (or bucket), and the soil is the slave surface. The sides of the soil are the ultimate lateral displacement, the bottom surface is fixed, and the top surface is free and unconstrained.

*2.4. Process and Result Processing*

After the calculation, in ABAQUS, the internal force, such as bending moment and shear force, can be outputted from a specified section. In this paper, 80 sections were selected on the pile, and the internal force was extracted after the analysis.

**3. Model Test Verification**

To verify the correctness of the numerical simulations, including model size, the choice of the constitutive model, and FEM meshing, the results from the model test are compared with the results from FEM. For a super large-diameter monopile for a wind farm of offshore wind turbines, Zhu B. et al. [20], based on the rigid pile with a similar prototype and model under constant gravity, used a pilot model test with a model scale of 1:30 in the foundation, which was made of silt mining from an excavation pit in Hangzhou City.

For a certain group of Zhu B's tests, the foundation *t* was located in the silt soil layer. The finite element model of soil was established by the M–C model, and the parameters are shown in Table 3.

**Table 3.** Calculation parameters for the silt.

| Soil Layer | Effective Unit Weight | Young's Modulus | Friction Angle | Dilation Angle | Cohesive | Poisson's Ratio |
| --- | --- | --- | --- | --- | --- | --- |
| | $\gamma$/kN/m$^3$ | *E*/MPa | $\varphi$/° | $\psi$/° | *c*/kPa | $v$ |
| Silt | 9.31 | 1.73 | 41.5 | 13.04 | 0.5 | 0.28 |

The three-dimensional finite element software ABAQUS is used for finite element numerical analysis to model and analyze the full-scale model. The whole foundation model is a rectangular parallelepiped. The length and width of the model are 20 *D*, and the bottom of the pile is 5 *D* from the bottom of the model. The parameters of the model pile are shown in Table 4.

**Table 4.** Parameters of the model pile.

| Diameter | Thickness | Buried Depth | Length | Density | Young's Modulus | Poisson's Ratio |
| --- | --- | --- | --- | --- | --- | --- |
| *D*/m | $\delta$/m | *h*/m | *l*/m | $\rho$/kg/m$^3$ | *E*/MPa | $v$ |
| 0.165 | 0.003 | 0.915 | 1.905 | 7800 | $2.1 \times 10^5$ | 0.3 |

Figure 4 compares the load and the displacement calculated from ABAQUS to the test results by Zhu B. The loading point heights are 1 *D*, 3 *D*, or 6 *D*. When the loading height is 1 *D*, the numerical result obtained from ABAQUS is well agreed with the results from Zhu B.'s [14] test for both loading phase and unloading phase. During the loading phase, the load–displacement curve is almost identical between the numerical simulations and the model test. When the loading height is 3 *D*, although there is a gap between the results of ABAQUS and Zhu B.'s test, the trend of load versus displacement curve is still consistent. When the loading height is 6 *D*, the agreements of the results are also good. Figure 5 presents the pile deflection curve under the load conditions of 1068, 1214 and 1367 N (the loading point height is 6 *D*). Similar agreements are observed from the comparison between numerical simulations and the model test.

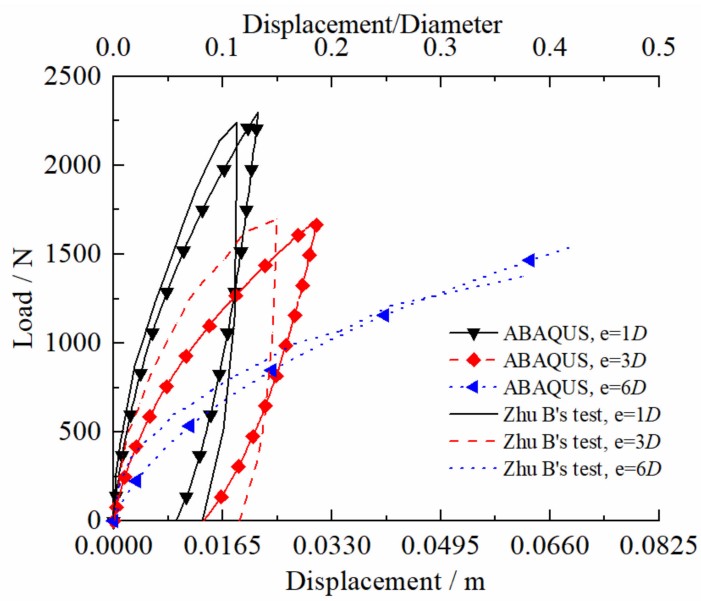

**Figure 4.** Load–displacement curves at the loading point.

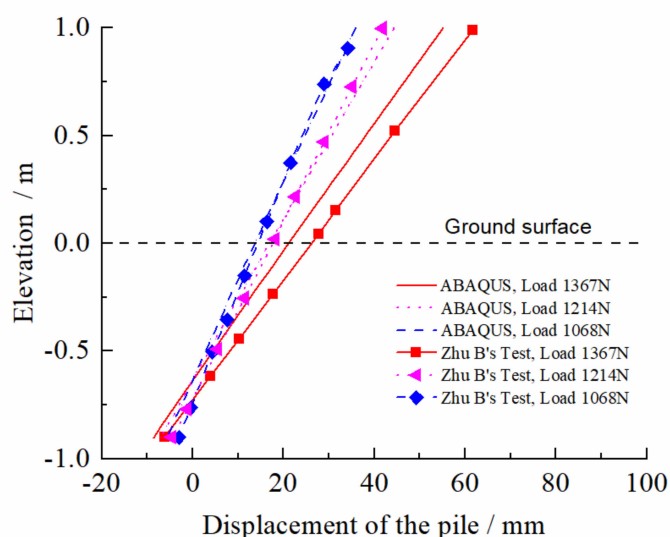

**Figure 5.** Pile deformation under various loading levels.

Based on the above analysis, we can conclude that the results of the simulation are close to those of Zhu B's tests, which can verify that the model parameters in the M–C model are effective, and the M–C model is quite reliable for static analysis.

## 4. Results of Numerical Analysis

### 4.1. Definition and Influence of Rigid–Flexible Pile

Under the action of lateral load, different pile foundations have different deformation characteristics (i.e., lateral displacement of pile) and failure behaviors. In foundation engineering, these are divided into rigid pile and flexible pile. There are many experts and scholars in the world who have studied and defined the discrimination methods and standards of rigid piles and flexible piles. Dobry R. [21] summarized the displacement response at the top of the pile through elastic foundation beam (BEF) and dynamic finite element analysis; thus, the stiffness or flexibility of the pile is not related to the length of the pile, but to the ratio of the Young's modulus of the pile and soil. Budhu M. and Davies T. [22], through the analysis of the ratio of pile–soil stiffness and the ratio of soil weight to strength, the comparison of piles of different sizes and different types of soil was carried out, and the judgment method of rigid piles and flexible piles was summarized. Poulos and Hull [23] adopted a set of dimensionless ratios between pile bending stiffness and soil stiffness to determine whether the pile is flexible or rigid. This method is also the most widely used and most recognized criteria at present. The method is as follows:

$$\frac{E_\text{P} I_\text{P}}{E_\text{s} l_\text{L}^4} > 0.208, \text{ Rigid pile}$$

$$\frac{E_\text{P} I_\text{P}}{E_\text{s} l_\text{L}^4} < 0.0025, \text{ Flexible pile}$$

where $E_\text{P} I_\text{P}$ refers to bending stiffness of the pile; $E_\text{s}$ refers to Young's modulus of soil; $l_\text{L}$ refers to the depth of the pile into the soil. According to the size of the large-diameter piles in the pile–bucket foundation selected in this paper, the above formula is used to calculate the ratio of the flexural stiffness of the pile to the soil stiffness, which is 0.1031, and which is between the rigid piles and flexible piles, belonging to rigid–flexible piles.

### 4.2. Criteria for Judging Lateral Bearing Capacity

According to the research of Achmus [24], in the analysis of the foundation for wind turbines, the displacement control of the foundation is transformed into the control of the

rotation angle at the mudline surface, and the rotation angle at the mudline surface of the pile foundation is controlled within 0.5°. Therefore, in this article, a lateral displacement of 0.17 m was applied to the top of the pile–bucket foundation, and the rotation angle at the mudline surface of the pile–bucket foundation was controlled at 0.5°. The horizontal force at this rotation angle is defined as the lateral bearing capacity of the pile–bucket foundation.

### 4.3. Comparison of Bearing Capacity and Displacement of PF, BF and P–BF

According to the above, the PF, BF and P–BF ($d = 6$ m, $l = 80$ m, $D = 18$ m, $h = 12$ m) are established, respectively, where $d$ is pile diameter, $l$ is pile length, $D$ is bucket diameter, and $h$ is the height of bucket. As shown in Figure 6, the lateral bearing of the P–BF is much higher than the SF and BF. The lateral bearing capacity of the BF base is lower than the PF. Under the same load, the displacement of the BF is much larger than that of the PF, as is shown in Figure 7. Thus, we can conclude that the P–BF has much better anti-overturning ability than the BF under the same lateral load and lateral bearing capacity than the PF.

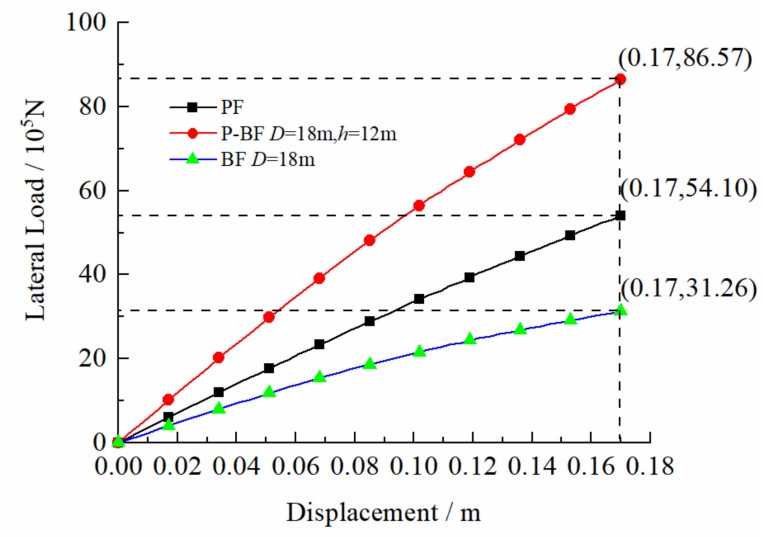

**Figure 6.** Comparison of the bearing capacity.

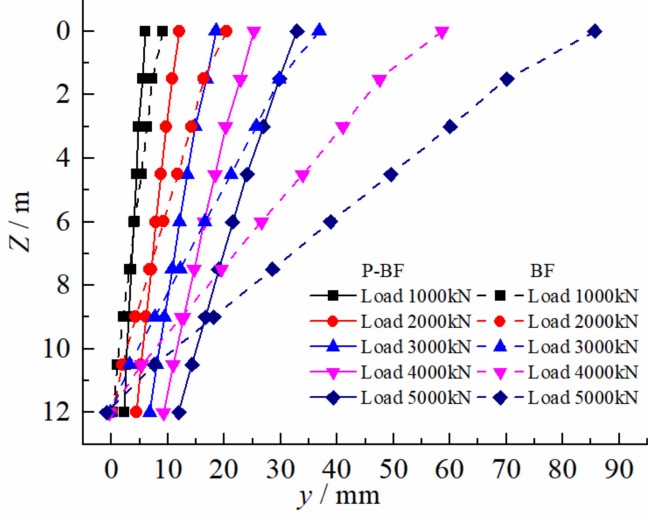

**Figure 7.** Comparison of lateral displacement between P–BF and BF.

### 4.4. Lateral Bearing Capacity of P–BF

In this paper, for comparison beside the P–BF model, a PF model is also established. Finally, the lateral displacement, shear force and moment of the foundation can be read from ABAQUS naturally.

Under the lateral load, the soil of around 4D depth provides the main stiffness of the foundation. In order to study the influence of the diameter ($D$) and height ($h$) of the bucket on the reinforcement, the height of the bucket is 1 $D$, 2 $D$ and 3 $D$, respectively. The bucket with a diameter of 3 $D$, 4 $D$ and 5 $D$ is adopted. The specific parameters are shown in Table 5.

**Table 5.** Parameters of model bucket.

|  | Model 1 | Model 2 | Model 3 | Model 4 | Model 5 |
|---|---|---|---|---|---|
| Diameter $D$/m | 18 | 24 | 24 | 24 | 30 |
| Height $h$/m | 12 | 6 | 12 | 18 | 12 |

Under the working conditions allowed by the specification, the soil around the foundation is still in the elastic domain. The bucket has a significant reinforcement effect on the foundation, and there are subtle differences in the different diameters and heights of the bucket for its reinforcement effect. After adding the bucket onto the PF, the lateral bearing capacity of the foundation is improved. Table 6 shows the improvement of lateral load capacity under different diameters and heights for the bucket. It can be obtained from Table 6 that the lateral bearing capacity can be increased by about 33.21% for each increase in the diameter of the bucket, and the lateral bearing capacity can be increased by about 21.43% for each increase in the height of the bucket. After comparative analysis, the effect of increasing the diameter of the bucket is 10% to 13% higher than that of increasing the height of the bucket.

**Table 6.** The increment of bearing capacity.

| Foundation | Bearing Capacity/kN | Improvement/% |
|---|---|---|
| PF ($D$ = 6 m) | 5409 | - |
| P–BF ($D/h$ = 18 m/12 m) | 8657 | 60.05 |
| P–BF ($D/h$ = 24 m/6 m) | 9212 | 70.31 |
| P–BF ($D/h$ = 24 m/12 m) | 10,460 | 93.38 |
| P–BF ($D/h$ = 24 m/18 m) | 11,530 | 113.16 |
| P–BF ($D/h$ = 30 m/12 m) | 12,250 | 126.47 |

Figure 8 shows the plastic zone under the normal working limit state of the monopile foundation and pile–bucket foundation (taking $D$ = 18 m, $h$ = 12 m foundation as an example). As shown in Figure 8a, the plastic zone of the monopile foundation is mainly distributed around the soil-facing surface of the pile side, and the depth of the plastic zone is within the range of one pile diameter. As shown in Figure 8b, the plastic zone of the pile–bucket foundation is also observed at the soil-facing surface of the foundation, but it is different from the monopile foundation case. The depth of the plastic zone on the soil-facing surface on the side of the bucket is about one pile diameter, while the depth of the plastic zone on the side of the pile is about the same as the bucket height ($h$).

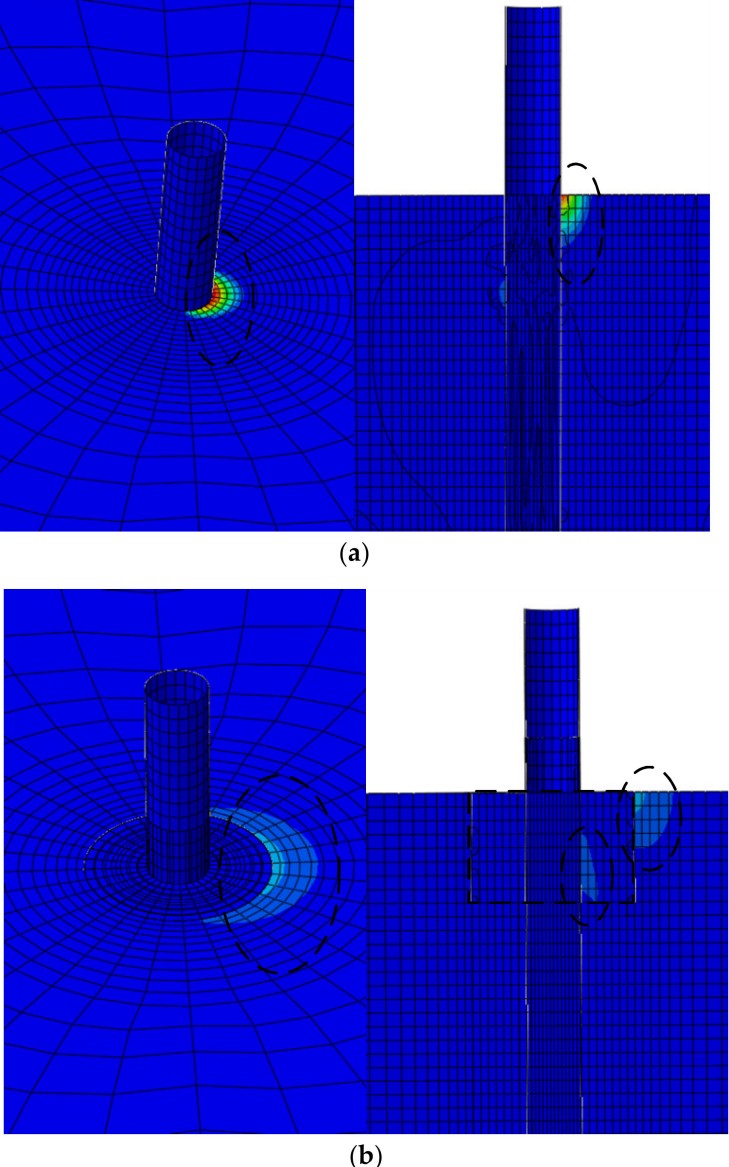

**Figure 8.** Plastic area distribution. (**a**) Monopile foundation. (**b**) Pile–bucket foundation.

*4.5. Comparison of P–BF Displacement and Soil Reaction Force*

4.5.1. The Displacement Distribution for Pile and Bucket

Figure 9 shows the displacement of the pile and the bucket.

Figure 10 shows the overall displacement of the PF and the P–BF under five loading conditions. a After the PF is strengthened by the bucket to form the P–BF, the rigidity of the foundation upper part is increased. Under the lateral load, the rotation center of the foundation is moved upward. Moreover, the relative stiffness of the P–BF with a depth of more than 20 m is reduced, and the deformation characteristic is closer to the flexible pile. In order to clearly see the increase in stiffness, Figure 11 shows the overall displacement of the PF and the P–BF of five sizes under 5000 kN loading.

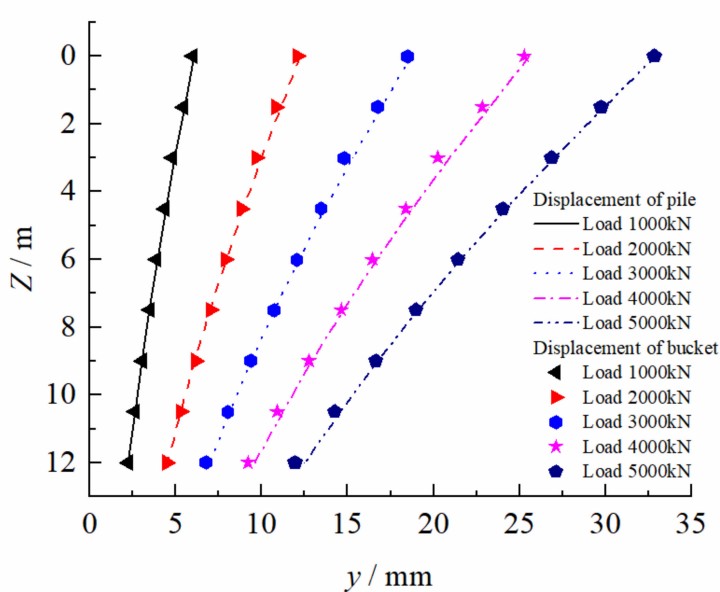

**Figure 9.** Lateral displacement distribution of bucket-reinforced part.

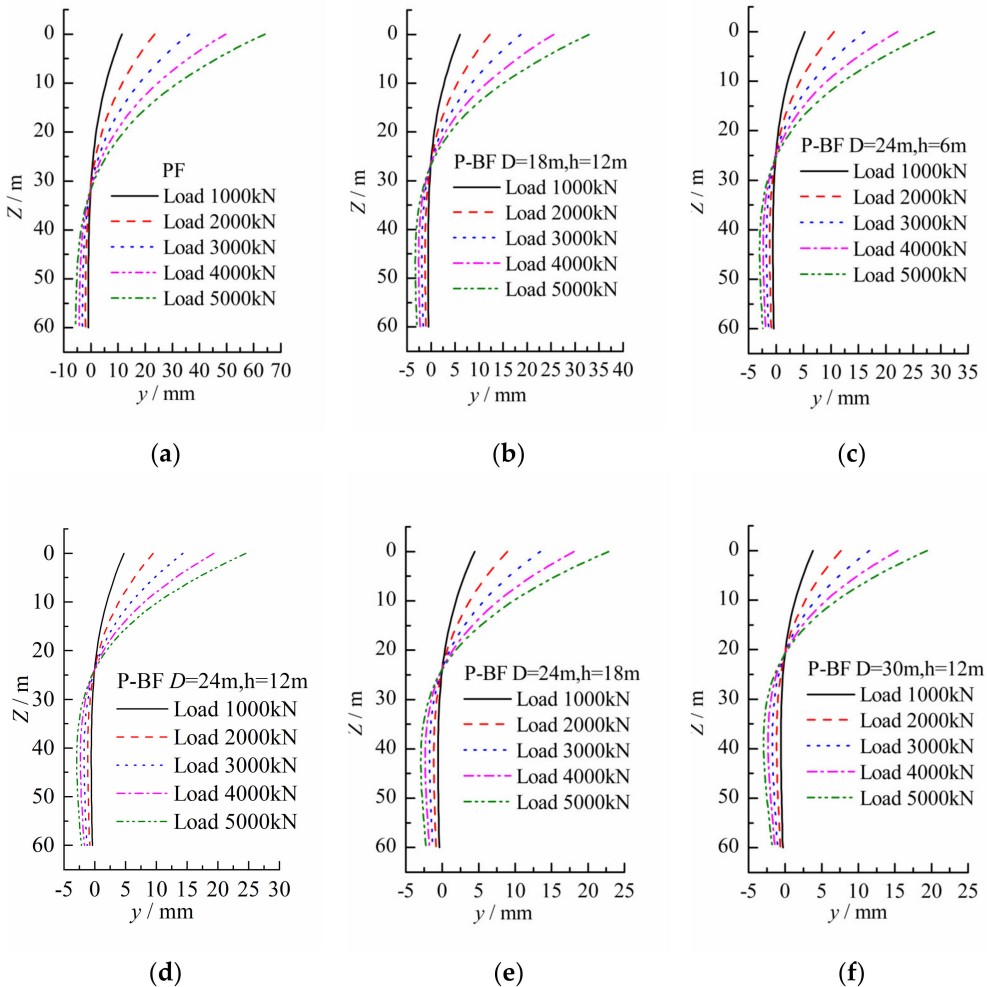

**Figure 10.** Lateral displacement of PF and P–BF. (**a**) Lateral displacement of PF. (**b**) Lateral displacement of P–BF. (**c**) Lateral displacement of P–BF; $D/h$ = 18 m/12 m $D/h$ = 24 m/6 m. (**d**) Lateral displacement of P–BF. (**e**) Lateral displacement of P–BF. (**f**) Lateral displacement of P–BF; $D/h$ = 24 m/12 m $D/h$ = 24 m/18 m $D/h$ = 30 m/12 m.

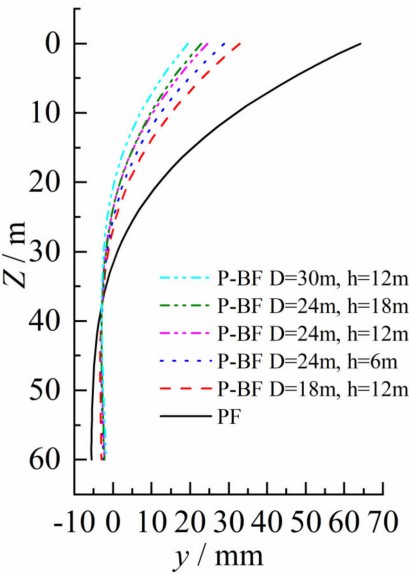

**Figure 11.** Lateral displacement of PF and P–BF under 5000 kN loading.

4.5.2. Distribution of Reactive Load

The reinforcement section of the P–BF will certainly bear a larger part of the lateral soil reaction force while the foundation stiffness increases. Figure 12 shows the distribution of the soil reaction under five loading conditions. In addition, in Figure 12, it is observed that the depth of the subgrade reaction turning point for P–BF is also different from that of PF. The depth of the subgrade reaction of the P–BF is almost the same as the height of the bucket.

Under these five different loading conditions, the bucket shares more than 90% of the soil reaction in the reinforcement part. Therefore, the main bearing member of the P–BF under lateral load is the bucket. Combining the above, the anti-overturning ability of the P–BF is superior to BF. It can be found that under the lateral load, the bucket in the P–BF provides lateral bearing capacity for the foundation, and the pile provides sufficient an anti-overturning moment.

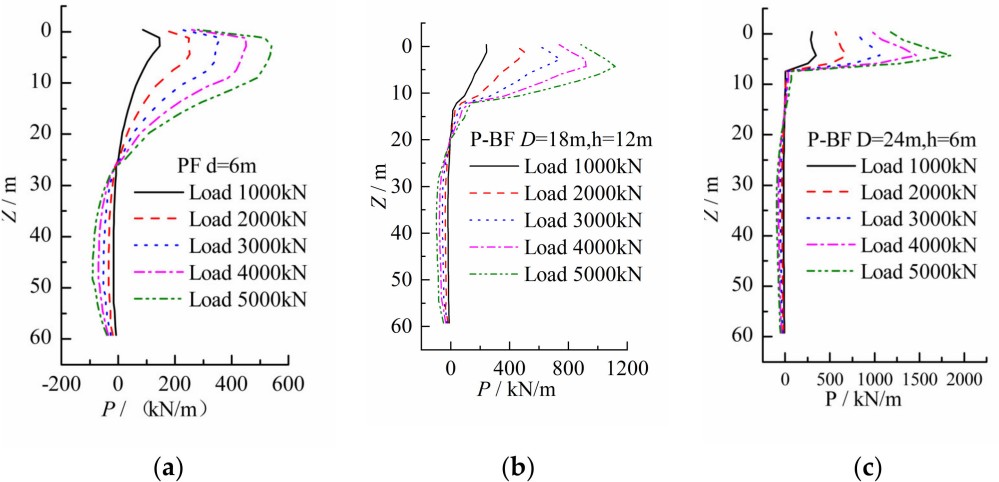

**Figure 12.** *Cont.*

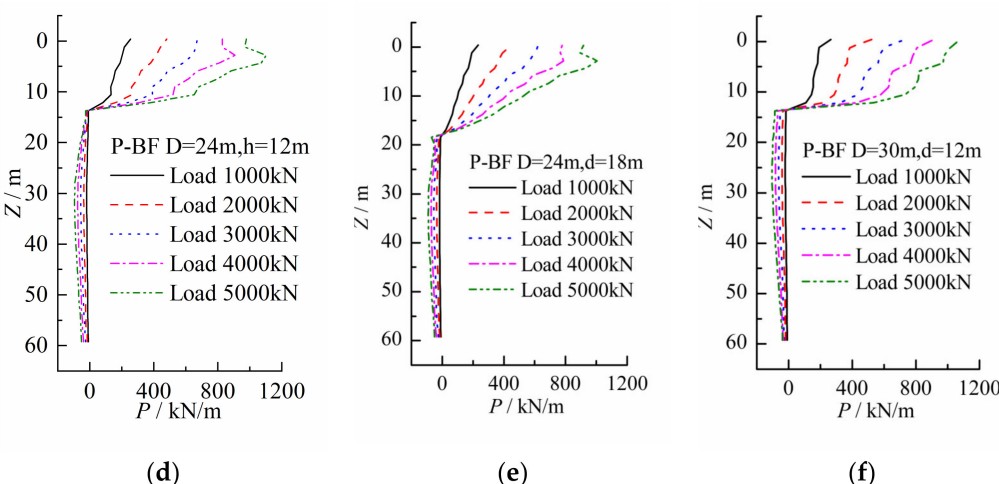

**Figure 12.** The subgrade reaction force distribution of PF and P–BF. (**a**) The subgrade reaction distribution. (**b**) The subgrade reaction distribution. (**c**) The subgrade reaction distribution of PF of P–BF; $D/h$ = 18 m/12 m $D/h$ = 24 m/6 m. (**d**) The subgrade reaction distribution. (**e**) The subgrade reaction distribution. (**f**) The subgrade reaction distribution of P–BF; $D/h$ = 24 m/12 m of P–BF, $D/h$ = 24 m/18 m of P–BF, $D/h$ = 30 m/12 m.

## 5. Analysis on *p-y* Curve

### 5.1. Applicability Analysis of API Specification p-y Curve Method

The *p-y* curve method is widely used in the analysis of lateral deformation of traditional PF and is adopted by the American Petroleum Institute (API). However, the design theory given by the API specification is based on a PF with a diameter less than 1.5 m, which is not suitable for large-diameter piles. In this subsection, the applicability of the API specification *p-y* curve to the large-diameter PF was studied by establishing a finite element numerical model with diameters of 2, 4 and 6 m and a thickness to diameter ratio of 1%. According to the empirical calculation, the piles with diameters of 2, 4 and 6 m are rigid–flexible piles, but the pile with a diameter of 2 m is close to the flexible pile. As is shown in Figure 13a, the *p-y* curve obtained according to the API specification is compared with the *p-y* curve obtained by the ABAQUS. At shallow depths, the lateral ultimate soil resistance of the soil from the API specification curve is somewhat different from the ABAQUS finite element simulation for the large-diameter pile with a diameter of 2 m. Taking a depth of 2.25 m as an example, the result of ABAQUS is nearly 1.5 times that of the API specification. For the diameter of 4 m, the result is shown in Figure 13b. For the large-diameter pile with a diameter of 6 m, the API specification seriously underestimates the lateral ultimate soil resistance of the soil at a depth of 1 *d*, but overestimates the lateral ultimate soil resistance of the soil at a shallow depth. As shown in Figure 13c, in the rising part of the curve, when the displacement is small, the curves of the same depth are well fitted, and the result is close. When the displacement is small and the soil is shallow, it is thus obtained that the curve recommended by the API specification still has certain usability in the PF with the pile's diameter as 6 m.

In summary, the *p-y* curve of the API specification for the large-diameter PF is close to the flexible pile, and the lateral ultimate soil resistance is underestimated in each layer of soil in relation to real behavior. For rigid or rigid–flexible large-diameter PF, there is an overestimate of lateral ultimate soil resistance in shallow soils; however, it is seriously underestimated in the deep soil. Thus, it can be concluded that the API specification is too conservative.

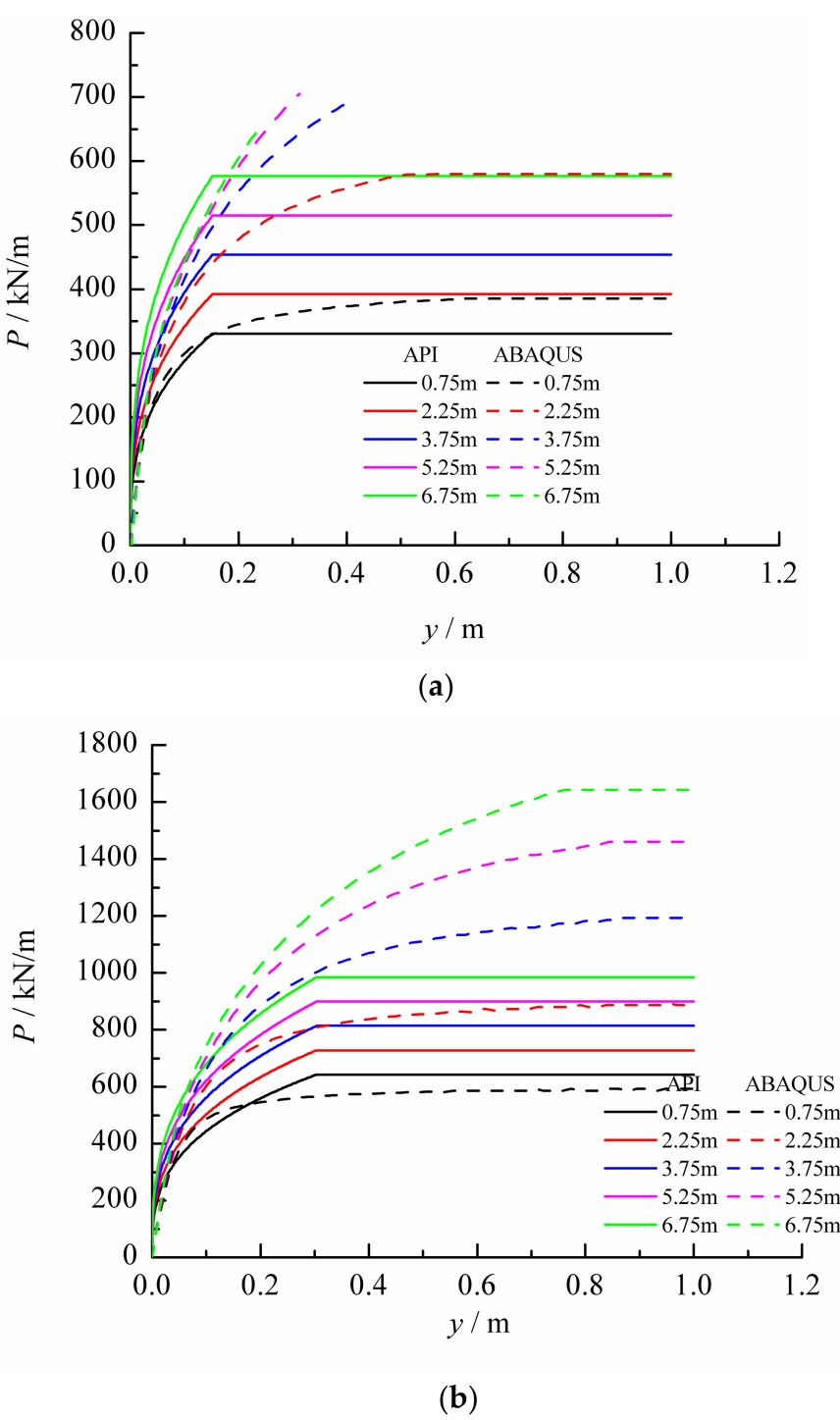

**Figure 13.** *Cont.*

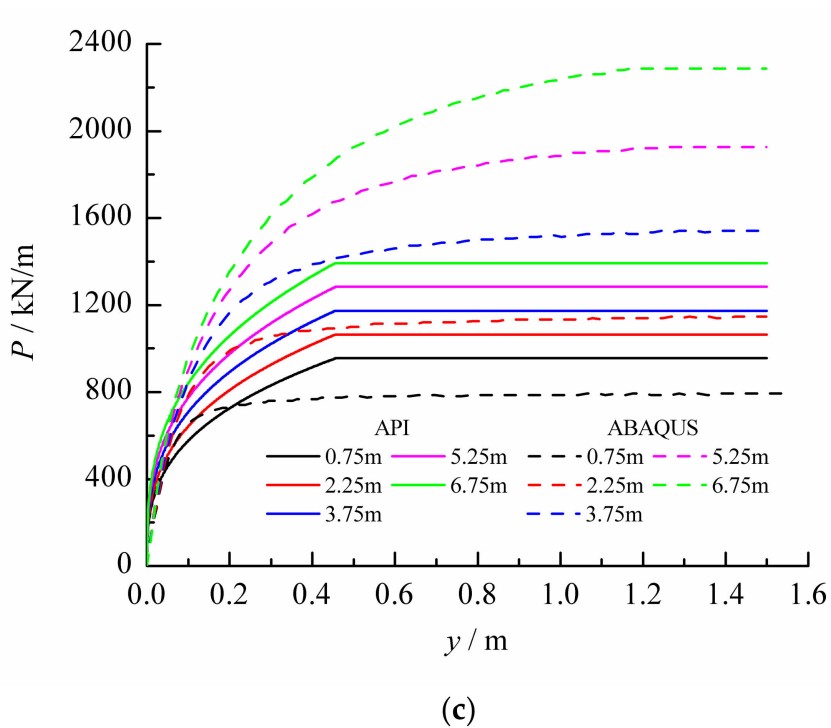

**(c)**

**Figure 13.** *p-y* curve of *D* = 2, 4, 6 m. (**a**) *p-y* curve of *D* = 2 m. (**b**) *p-y* curve of *D* = 4 m. (**c**) *p-y* curve of *D* = 6 m.

*5.2. Comparison of p-y Curves between PF and P–BF*

It has been proven above that the bucket greatly improves the lateral bearing capacity of the PF, and the *p-y* curve recommended by API specification seriously underestimates the lateral ultimate soil resistance of the soil below the 1 *d* of the large-diameter PF, which is too conservative. In order to study the reinforcement of the foundation for rigid–flexible piles, a large-diameter pile with a diameter of 6 m is used in the following calculations. Figure 14 shows the *p-y* curve of the P–BF. Because of the difference in bucket height, the *p-y* curve of the P–BF with depths of 2.25 and 3.75 m is compared as an example. In the same depth of shallow soil, the lateral ultimate soil resistance of the soil can be increased to more than seven times after the reinforcement of the bucket. At the same time, it has been found that increasing the height of the bucket has a limited improvement in the lateral ultimate soil resistance at a depth, as is shown in Table 7a,b. Moreover, the bucket of the same diameter increases with the height of the reinforced bucket, and the lateral ultimate soil resistance at the same depth is slightly lower. The lateral resistance of the whole foundation increases with the increase in bucket height.

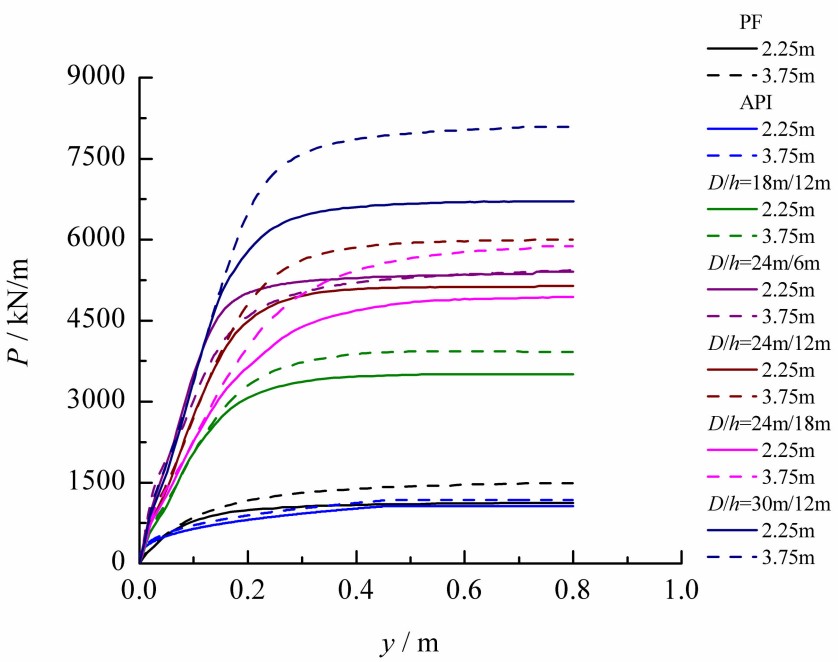

**Figure 14.** *p-y* curve of the pile–bucket foundation.

**Table 7.** (a) The increment of $P_{ult}$ at a depth of 2.25 m; (b) The increment of $P_{ult}$ at a depth of 3.75 m.

| Model | Lateral Ultimate Soil Resistance $P_{ult}$/kN/m | Improvement/% |
|---|---|---|
| **(a)** | | |
| API | 1065 | - |
| PF ($d$ = 6 m) | 1150 | - |
| P–BF ($D/h$ = 18 m/12 m) | 3426 | 197.91 |
| P–BF ($D/h$ = 24 m/6 m) | 5406 | 370.09 |
| P–BF ($D/h$ = 24 m/12 m) | 5140 | 346.96 |
| P–BF ($D/h$ = 24 m/18 m) | 4940 | 329.57 |
| P–BF ($D/h$ = 30 m/12 m) | 6700 | 482.61 |
| **(b)** | | |
| API | 1173 | - |
| PF ($d$ = 6 m) | 1540 | - |
| P–BF ($D/h$ = 18 m/12 m) | 3846 | 149.74 |
| P–BF ($D/h$ = 24 m/6 m) | 5433 | 252.79 |
| P–BF ($D/h$ = 24 m/12 m) | 6000 | 289.61 |
| P–BF ($D/h$ = 24 m/18 m) | 5880 | 281.82 |
| P–BF ($D/h$ = 30 m/12 m) | 8133 | 428.12 |

### 5.3. Modification of p-y Curves for P–BF

The displacement, the distribution of the subgrade reaction, and the form of the *p-y* curve of the P–BF are similar to those of the PF, according to the *p-y* curve of the PF, recommended by the API specification, and modifying those of P–BF. Under the static force, the *p-y* curve expression in clay recommended by the API specification is expressed by (1).

$$\begin{cases} P = \frac{P_{ult}}{2}\left(\frac{y}{y_{50}}\right)^{1/3}, y \le 8y_{50} \\ P = P_{ult}, y \le 8y_{50} \\ y_{50} = 2.5\varepsilon_{50}d \end{cases} \tag{1}$$

where

$P$ = actual lateral resistance, kN/m;

$P_{ult}$ = ultimate resistance, kN/m;

$y$ = actual lateral deflection, m;

$y_{50}$ = corresponding lateral deflection of the half lateral ultimate soil reaction force, m;

$\varepsilon_{50}$ = strain that occurs at one-half the maximum stress on laboratory unconsolidated undrained compression tests of undisturbed soil samples;

$d$ = pile diameter, m;

$P_{ult}$ resistance (in the API specification) under static load is calculated as

$$
\begin{cases}
P_{ult} = (3s_u + \gamma Z)d + Js_u Z, 0 < Z \le Z_r \\
P_{ult} = 9s_u d, Z > Z_r \\
Z_r = \frac{6s_u d}{\gamma + Js_u}
\end{cases}
\tag{2}
$$

where $S_u$ = undrained shear strength for undisturbed clay soil samples, kPa;

$\gamma$ = effective unit weight of soil, kN/m$^3$;

$J$ = dimensionless empirical constant with values ranging from 0.25 to 0.5 having been determined by field testing; a value of 0.5 is appropriate for normally consolidated clay;

$Z_r$ = depth below soil surface to bottom of reduced resistance zone, m.

The point of first intersection of the two equations in Formula (2) is taken to be $Z_r$. These empirical relationships may not apply where strength variations are erratic. In general, the minimum value of Zr should be about 2.5 pile diameters.

The remaining parameters are consistent with the above.

It can be obtained for (1) and (2) that $P_{ult}$, $y_{50}$, $P$ and $y$ are the key parameters for modifying the *p-y* curve of the P–BF. Combined with the results of ABAQUS, the *p-y* curve of the P–BF can be obtained by modifying the above parameters with reference to the method of large-diameter winged pile *p-y* curve modification mentioned by Hu Y [25] et al.

For the *p-y* curve at different depths of the P–BF obtained by ABAQUS simulations, ($d$ = 6 m $h$ = 12 m), the ultimate resistance of the foundation is the platform section in the *p-y* curve. The ultimate resistance of the P–BF was calculated and is recorded in Table 8. The ultimate resistance of the P–BF is significantly increased with an increasing depth, and it is significantly higher than that of the PF.

**Table 8.** Lateral ultimate soil reaction.

| d/m | D/m | Depth Z/m 2.25 | 3.75 | 5.25 |
|-----|-----|------|------|------|
|  |  | $P_{ult}$/kN/m | | |
| 3 | 18 | 3767 | 4273 | 4425 |
| 4 | 18 | 3747 | 4213 | 4536 |
| 5 | 18 | 3660 | 4133 | 4527 |
| 6 | 18 | 3527 | 3847 | 4507 |
|  | 24 | 5140 | 6000 | 6407 |
|  | 30 | 6700 | 8133 | 8953 |
|  | API | 1064 | 1174 | 1283 |

At the same time, a model was built to determine the relationship between the pile diameter and the reinforcement bucket diameter. The parameters are chosen as follows: diameter–thickness ratio 1%, $h$ is 12 m, $D$ is 18 m. We obtained the *p-y* curves in the P–BF reinforcement section at different depths, which is shown in Figure 15. It indicates that different pile diameters have little effect on the lateral ultimate soil reaction of the pile body in the reinforcement section of the P–BF. The effect decreases as the depth increases, even if there is no change in the diameter or height of the pile. Therefore, in this paper, the influence of the pile diameter is not considered while modifying the *p-y* curve.

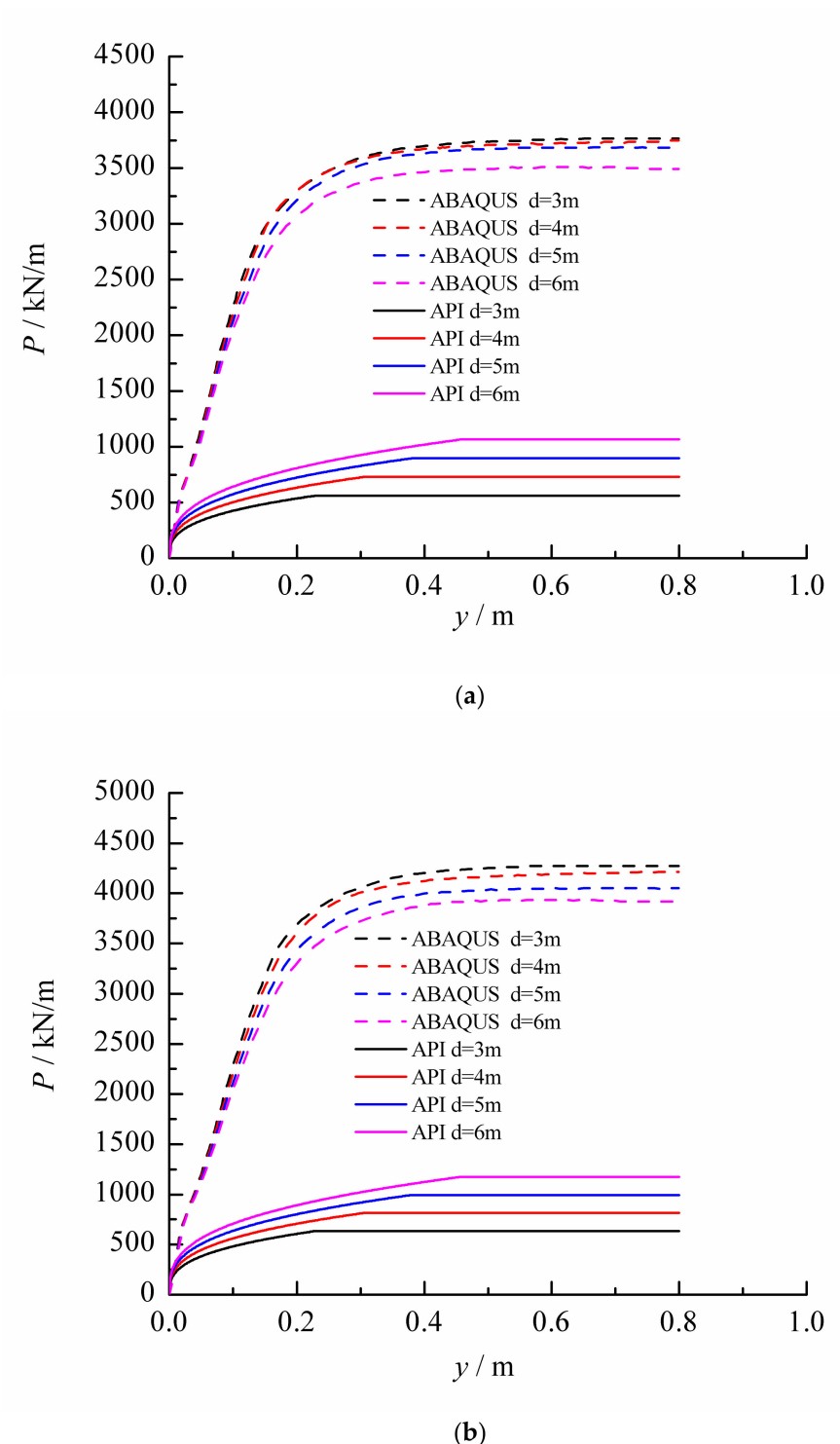

**Figure 15.** *Cont.*

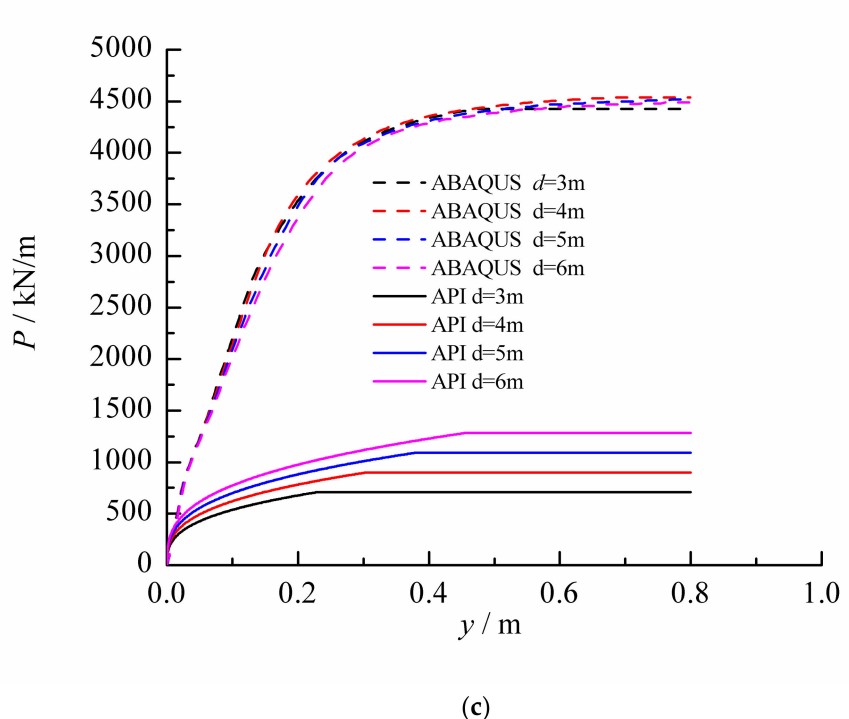

(c)

**Figure 15.** *p-y* curve at different depths. (**a**) Depth of 2.25 m. (**b**) Depth of 3.75 m. (**c**) Depth of 5.25 m.

5.3.1. Modification of Lateral Ultimate Soil Reaction

According to Equation (2), when the pile is above the turning point of the lateral limit soil reaction, the lateral ultimate soil reaction of piles is affected by undrained shear strength $S_u$, soil weight $\gamma$, depth $Z$ and pile diameter $d$. As verified above, the diameter of the pile in P–BF has little effect on the reinforcement section in the *p-y* curve; thus, the $d$ in P–BF can be considered to be the same as the diameter of the bucket. When the soil parameters and the calculated width are known, the API specification recommends that the lateral ultimate soil resistance in the curve can be transformed into a one-variable function related to depth $Z$. Equation (3) is obtained by adding a modification coefficient to Equation (2).

$$\begin{cases} P_{ult} = (es_u + f\gamma Z)d_s + gs_u Z \\ d_s = \alpha d \\ \alpha = \frac{D}{d} \end{cases} \tag{3}$$

where

$d_s$ = calculated width, m;

$D$ = Bucket diameter, m;

Other parameters are the same as above.

Qian J H [26] et al. found that the calculation of geotechnical problems using the M–C failure model within the limited stress range is in good agreement with the measured results. Above the turning point of the lateral ultimate soil reaction, the stress is relatively small. Thus, the undrained shear strength above the turning point is calculated by the shear strength formula, which has a linear relationship with depth.

$$s_u = c_u + \gamma Z \tan \varphi \tag{4}$$

The parameters in the formula are consistent with the above.

When Formula (4) is inserted into Formula (3), the following expression is obtained.

$$\begin{cases} P_{ult} = A + BZ + CZ^2 \\ A = ecd_s \\ B = e\gamma d_s tan\varphi \\ C = g\gamma tan\varphi \end{cases} \tag{5}$$

According to the results of the soil reaction distribution, the turning point of the lateral limit soil reaction should be below the wall of the bucket in engineering design. Under the same load, the distribution of the soil reaction outside the bucket section of the P–BF bucket agrees well with that of PF [27]. Based on the results of the ABAQUS simulation, the undetermined coefficients *A*, *B* and *C* in Formula (5) are solved by the least square method. Then, the formulas of *e*, *f* and *g* are inversely solved, and the formula of the lateral ultimate soil reaction above the turning point is established.

$$\begin{cases} P_{ult} = (9.059s_u + 1.605\gamma Z)d_s - 10.402s_u Z , Z \leq Z_r \\ P_{ult} = 9s_u , Z > Z_r \end{cases} \tag{6}$$

Other parameters are the same as above.

The revised results are consistent with the fitting trend of the finite element simulation results, which are shown in Figure 16.

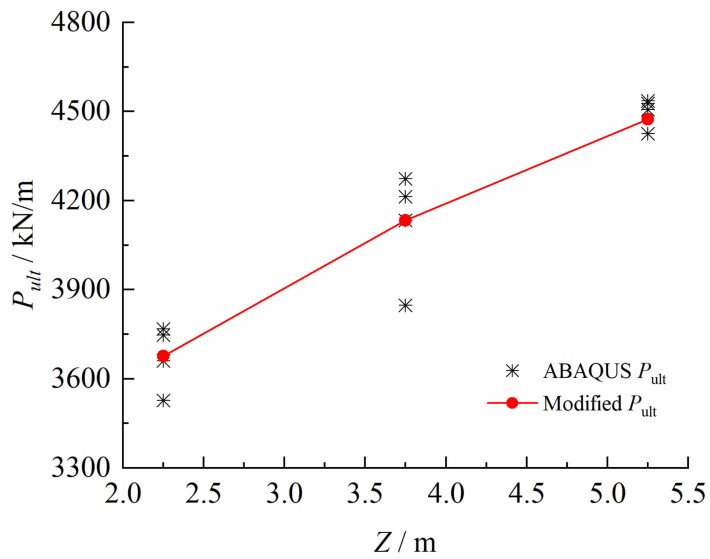

**Figure 16.** Modified lateral ultimate soil reaction.

5.3.2. Modified Eigenvalue $y_{50}$ of Pile Displacement

The eigenvalue of pile displacement is another important parameter of the *p-y* curve, which is referred to as the eigenvalue of the displacement of the pile. It is obtained from the *p-y* curve simulated by ABAQUS as shown in Table 9. There is a linear relationship between $y_{50}$ and depth *Z*. Then, a dimensionless coefficient k, which is related to depth *Z*, is introduced to reconstruct $y_{50}$, as shown in Formula (7).

$$\begin{cases} y_{50} = kd_s \\ k = m + n(Z/h) \end{cases} \tag{7}$$

**Table 9.** Displacement of the pile.

| Depth Z/m | | 2.25 | 3.75 | 5.25 |
|---|---|---|---|---|
| *d*/m | *D*/m | $y_{50}$/m | | |
| 3 | 18 | 0.07920 | 0.10430 | 0.09713 |
| 4 | 18 | 0.07559 | 0.09622 | 0.10450 |
| 5 | 18 | 0.08069 | 0.09894 | 0.10996 |
| | 18 | 0.08572 | 0.09659 | 0.11423 |
| 6 | 24 | 0.09077 | 0.11274 | 0.12687 |
| | 30 | 0.09961 | 0.11962 | 0.12699 |
| | API | | 0.05700 | |

The undetermined coefficients m and n in Formula (7) are solved by the least squares method. Formula (8) is the revised formula of $y_{50}$.

$$y_{50} = [0.0291(Z/h) + 0.0035]d_s \qquad (8)$$

where

  *h* = the bucket height, m;

  Other parameters are the same as above.

  The revised eigenvalue is consistent with the fitting trend of the finite element simulation results, which are shown in Figure 17.

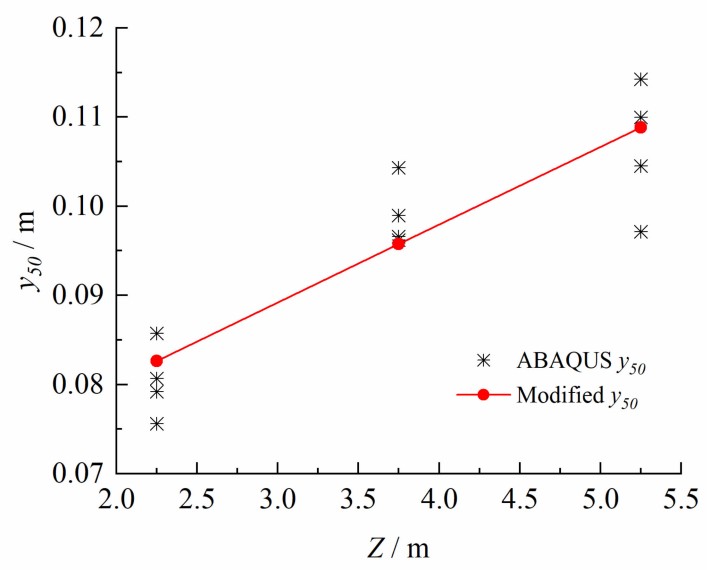

**Figure 17.** Modified displacement of the pile.

### 5.3.3. Modified $P/P_{ult}$-$y/y_{50}$ Curves

The dimensionless *p-y* curve ($P/P_{ult}$-$y/y_{50}$) is adopted in API to reflect the *p-y* curve of PF under a lateral load. This curve form can reflect the relationship between soil reaction and lateral displacement of PF. Modifying $P/P_{ult}$-$y/y_{50}$ according to the formula given in API is an effective way to quantitatively analyze the *p-y* curve of P–BF. The *p-y* values of P–BF obtained by the finite element simulation are calculated and plotted in a dimensionless way as $P/P_{ult}$-$y/y_{50}$ curves (see Figure 18). It can be found that the $P/P_{ult}$-$y/y_{50}$ of the bucket of P–BF is basically the same as the recommended curve of API, but there are also some differences. Before reaching the lateral ultimate soil reaction, the results of the finite element simulation and API curves are typical power function distributions. However, when the API curve reaches the lateral ultimate soil reaction, the $y/y_{50}$ value is significantly

higher than the results of the finite element simulation, which leads to a great gap between the secant slope of the recommended curve in the API code and the results of the finite element simulation. Nevertheless, the curves of the current code can still be applied to the P–BF after being modified.

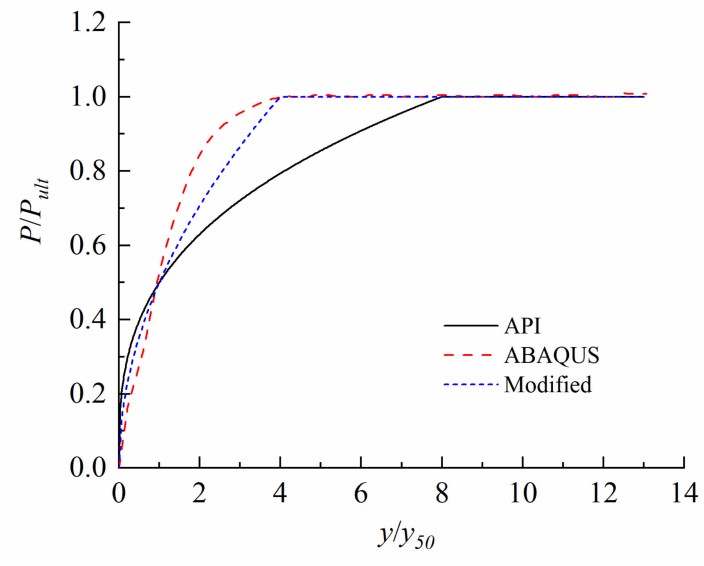

**Figure 18.** $P/P_{ult}$-$y/y_{50}$ curve.

If we transpose Formula (1) and rewrite the coefficient, we can obtain:

$$P/P_{ult} = a(y/y_{50})^{\frac{1}{b}} \tag{9}$$

When the eigenvalue of the pile displacement $y_{50}$ is half of the ultimate lateral soil reaction, the corresponding lateral pile displacement is as follows. That is, $y = y_{50}$, $P = 1/2$ $P_{ult}$, $a = 0.5$, which then yields

$$P/P_{ult} = 0.5(y/y_{50})^{\frac{1}{b}} \tag{10}$$

Applying a logarithm on both sides of Equation (10) gives

$$\ln(P/P_{ult}) = \ln 0.5 + \frac{1}{b}\ln(y/y_{50}) \tag{11}$$

Based on the ordinal ($y/y_{50}$, $P/P_{ult}$) data to calculate [ln ($y/y_{50}$, ln ($P/P_{ult}$)], we use the numerical approximation method to obtain the undetermined coefficient $b$. Thus, the *p-y* curve formula, which is suitable for P–BF, is obtained:

$$\begin{cases} P = \frac{P_{ult}}{2}\left(\frac{y}{y_{50}}\right)^{1/2}, y \leq 3.99 y_{50} \\ P = P_{ult}, y > 3.99 y_{50} \end{cases} \tag{12}$$

The other parameters are the same as above.

The $P/P_{ult}$-$y/y_{50}$ curve is calculated through the revised *p-y* curve formula, and the *p-y* curve at the depths of 2.25 and 3.75 m of the P–BF (taking the four groups of models with a diameter of 18 m as examples) is compared with the results of the ABAQUS finite element simulation, as shown in Figures 18 and 19. It can be seen from the graph that the modified *p-y* curve has satisfactory results. It can basically reflect the relationship between the lateral soil reaction of P–BF and the lateral displacement of the pile under lateral load.

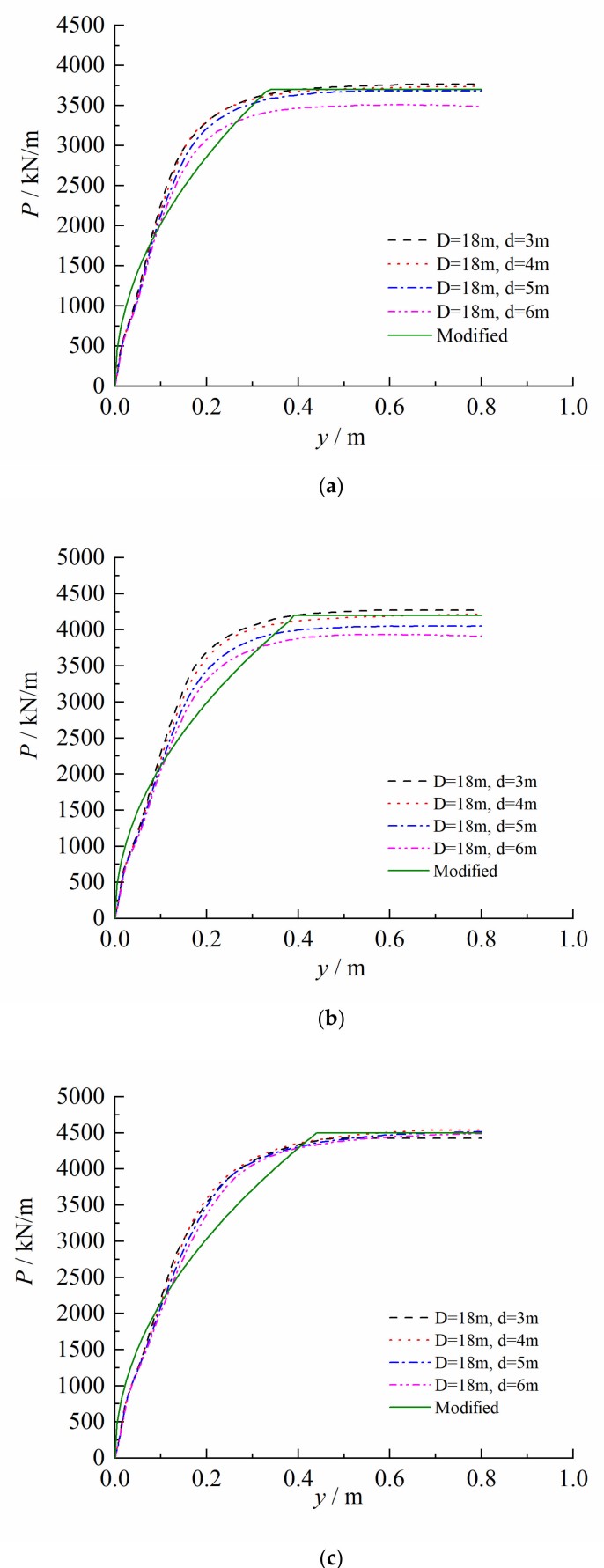

**Figure 19.** Modified *p-y* curve. (**a**) Depth of 2.25 m. (**b**) Depth of 3.75 m. (**c**) Depth of 5.25 m.

## 6. Conclusions

In this paper, the finite element analysis model of P–BF with different pile diameter/bucket diameter combinations was established in ABAQUS to analyze the effect of a tube on the lateral bearing performance of a large-diameter single pile. The results demonstrate that in P–BF, the bucket mainly provides lateral bearing resistance and the pile mainly provides an anti-overturning moment. Compared with the height of the bucket, the diameter of the bucket has a greater effect on the lateral bearing resistance of the pile–bucket foundation. In particular, based on the results of the finite element simulation, the API specification recommends a monopile revised *p-y* curve, which is obtained and which is suitable for the composite foundation level load performance of the pile *p-y* curve.

The conclusions are as follows:

(1) The internal force distribution of P–BF is consistent with the rule of PF. The reinforcement effect of the bucket on the pile body is significant. In order to enhance the bearing capacity of P–BF, it is more effective to increase the diameter of the bucket body at the same magnitude.

(2) Within the reinforced range of the bucket, the lateral ultimate soil reaction of the P–BF at the same depth is much larger than that of the PF. The eigenvalue $y_{50}$ of the pile displacement varies linearly with depth.

(3) The height of P–BF with the same diameter has a greater influence on the *p-y* curve than that of the shallow layer, and increasing the height of the pile–bucket reduces the $P_{ult}$ value to a certain extent. Due to the fact that the increase in bucket height does not improve the bearing capacity of P–BF, we conclude that when the pile diameter is greater than 6 m, increasing the bucket height can make the reinforced section of the foundation bear more uniform load on the basis of a constant bucket diameter.

(4) In the P–BF, the reinforcement section of the bucket is the main contribution to the lateral bearing capacity of the foundation, which provides more than 90% of the bearing capacity. At the same time, the unreinforced section in the pile provides enough anti-overturning capacity for the foundation, which significantly improves the overall lateral bearing capacity of the foundation.

(5) The distribution of the *p-y* curve of the P–BF is similar to that of PF. The *p-y* curve of PF recommended by API and the *p-y* curve of the finite element simulation of P–BF are typical power function distributions. However, when it is extended to P–BF, the key parameters need to be revised, and the revised methods and ideas proposed in this paper can be referred to.

**Author Contributions:** Conceptualization, Z.Y.; methodology, Z.Y. and X.P.; software, X.W. and J.W.; validation, X.P. and J.Z.; formal analysis, Z.Y. and B.H.; investigation, K.Z.; resources, B.H.; data curation, J.W.; writing—original draft preparation, K.Z. and X.W.; writing—review and editing, K.Z., J.Z. and S.X.; visualization, B.H.; supervision, X.P.; project administration, Z.Y.; funding acquisition, X.W. All authors have read and agreed to the published version of the manuscript.

**Funding:** This research was funded by National Natural Science Foundation of China grant number [51909249].

**Institutional Review Board Statement:** The study did not require ethical approval.

**Informed Consent Statement:** The study did not involve humans.

**Data Availability Statement:** No data were used to support this study.

**Conflicts of Interest:** The authors declare no conflict of interest.

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
