# Peer review of "A Numerical Analysis on Lateral Resistance of Pile–Bucket Foundation for Offshore Wind Turbines"

_applsci, doi:10.3390/app12094734_

Round 1
Reviewer 1 Report
The introduction needs improvement.Please check the first sentence of the introduction (line 26) as well as line 38. These passages seem to be nonsense.
The second paragraph seems to be lost in the introduction.
This excerpt (lines 39-48) should be removed for another section of the article.
Despite the citations in lines 58-72, the authors need to better connect this part of the article to the justification.
Reviewer 2 Report
Dear authors,
my comments are in the attached file.
Best regards

Reviewer 3 Report
The paper presents some investigatation by numeric modeling for marine wind turbine foundations.
The topic is actual, the paper could be very interesting, but:
- not sure the template was followed, layout is not looking too well, table 2 (line 148) break on multi pages, line 201 - 205 space etc)
- equation at line 102 is missing?
- section 2.3 very short
- only 21 references is very few, also majority are chinese authors. No one else in the world studies this field? Should be added to intro in state of the art, the status of the research worldwide, reviewed
- generally i would like to have actual ABACUS images inserted in the paper, with the deformations and models etc, not only the charts and graphs, but the developed models and their behaviour, to add some weight to the paper
- figure 4 and on, text in figures legend is larger than the actual text in the paper, looks bad.
